# Low Mediterranean Diet Adherence Is Associated with Poor Socioeconomic Status and Quality of Life: A Cross-Sectional Analysis

**DOI:** 10.3390/nu17050906

**Published:** 2025-03-05

**Authors:** Carolina Duarte, Andrea Campos, Telmo Pereira, João P. M. Lima

**Affiliations:** 1Coimbra Health School (ESTeSC), Polytechnic University of Coimbra, 3046-854 Coimbra, Portugal; carolinaisduarte@hotmail.com; 2Caritas Diocesana de Coimbra, 3030-382 Coimbra, Portugal; andreacampos@caritascoimbra.pt; 3H&TRC—Health & Technology Research Center, Coimbra Health School, Polytechnic University of Coimbra, 3045-043 Coimbra, Portugal; joao.lima@estesc.ipc.pt; 4GreenUPorto—Sustainable Agrifood Production Research Centre, 4169-007 Porto, Portugal; 5SUScita—Research Group on Sustainability, Cities and Urban Intelligence, 3030-199 Coimbra, Portugal

**Keywords:** quality of life, mediterranean diet, education, socioeconomic disadvantaged groups, Portugal

## Abstract

Background: Health-related quality of life (HRQoL) can be impacted by various environmental factors: lifestyle habits, food insecurity, social–economic status, and dietary patterns. The Mediterranean diet (MedDiet) has been associated with a healthier lifestyle and better health outcomes. The aim of this study was to determine whether greater adherence to the MedDiet was associated with better HRQoL in communities with low social and economic statuses living in two social neighborhoods, “Bairro da Rosa” and “Ingote”, in Portugal. Methods: A cross-sectional analysis was performed on 102 citizens currently receiving government social support and attending the “Health Kiosk”, a community center created within the Europe Enabling Smart Healthy Age-Friendly Environments (EU_SHAFE) project enhancing educative sessions and screening by health professionals. The participants answered a 36-item questionnaire about their HRQoL (SF-36) and a 14-item questionnaire about their adherence to the Mediterranean diet (MEDAS). Spearman correlation analysis between variables and multiple linear regression models were used to estimate the effect of the baseline characteristics and MedDiet adherence on HRQoL scores (SF-36). Results: No statistically significant correlation was found between MedDiet adherence and total HRQoL scores across the eight health concepts. However, age was shown to negatively influence HRQoL, whereas daily physical activity had a positive impact on health. SF-36 physical health concepts exhibited a linear trend with respect to MedDiet adherence, while emotional health concepts showed inconsistent patterns across adherence groups. Education and waist circumference influence HRQoL, with higher levels of education correlating with better quality of life and greater waist circumference being negatively associated with aspects such as energy and vitality. These findings suggest that factors beyond dietary patterns, such as physical activity, education, and body composition, play pivotal roles in shaping HRQoL in disadvantaged communities, emphasizing the need for multifaceted public health interventions.

## 1. Introduction

Inadequate eating habits represent the fifth risk factor (15.8%) that contributes the most to the total number of years of life lost in the Portuguese population [1]. In Portugal, more than 50% of the population is considered overweight and to have a high burden of chronic diseases, for which the adoption of a diet rich in sodium and processed meats and low in fruit, vegetables, whole grains, and nuts, combined with a lack of physical activity, excessive consumption of alcohol, and smoking habits, is a major contributor [1].

Several studies have shown that populations with higher levels of education and a better economic status are more likely to consume healthier foods [2] and have high rates of adherence to the MedDiet [3,4,5]. Despite the existent link between MedDiet adherence and the improvement of general health during one’s lifespan, only a few studies have considered the relation between this dietary pattern and HRQoL scales. As science has evolved, particularly in the field of health promotion and disease research, HRQoL domains have been a topic of growing interest. We now observe people living longer than our ancestors; however, these years are spent with a reduced quality of life [6,7]. The MedDiet, mostly adhered to in countries in the Mediterranean region, is considered one of the healthiest and most beneficial dietary patterns in the world. It has been linked since the 1960s with a lower incidence of diet-related chronic diseases [8] as well as with improvements in cognitive, metabolic, and cardiovascular conditions [9,10,11] and higher life expectancy [7,12]. These health outcomes are associated with the known anti-inflammatory and antioxidant properties of the foods herein referred to as plant-based foods (cereals, potatoes, beans, nuts and seeds, vegetables, and fruit), the daily use of olive oil as the primary source of fat, a moderate consumption of dairy or fish, and a low intake of red and processed meat and sweets/pastries. More than a dietary pattern, the MedDiet is a lifestyle, as it promotes sustainable and healthy environments, active and social interaction at the table with family and friends, and regular physical activity [8]. However, adherence to the MedDiet has been decreasing in southern European countries in the last few decades, especially among low-socioeconomic-status groups and younger groups. Having higher levels of education and income seems to be an advantage in terms of adherence to this dietary pattern. In a 2020 study [13] about adherence to the Mediterranean diet in Portugal, only half of the respondents were fully aware of the Mediterranean diet recommendations.

In Portugal, compared to previous decades, adherence to this diet has slightly increased since 2015 but is still well under the expected rate, with only 26% of Portuguese people having high adherence to the MedDiet. The guidelines for fruit, vegetable, and legume intake seem to be the hardest ones to follow due to unfamiliarity in regard to cooking or a reported dislike of taste among family members for these types of foods. Also, some myths around their healthy characteristics and a lack of knowledge on how to cook them further explain this trend. On the other hand, the underconsumption of fish or olive oil in the right amounts seems to be due to the fact that these foods are overpriced [14]. In recent research, a possible bidirectional relationship concerning the influence of a healthy diet on quality of life has been pointed out [15,16]. HRQoL has been shown to be a complex concept to define as it encompasses different domains of health and well-being, e.g., physical functioning, social and emotional problems, energy and fatigue, and health in general, and different research uses different criteria [17]. On the other hand, it can be a subjective topic and highly related to an individual’s self-perception of whether they feel their life is comfortable and satisfying. Such an evaluation can vary greatly between persons and cannot be excluded from clinical assessments and biological analyses.

For this reason, self-reported quality of life questionnaires have been considered some of the more accurate tools used in public health studies to evaluate such subjective matters. These types of questionnaires can also be used to decide on food policy guidelines and help design interventions.

The Mediterranean diet, recognized for its health benefits, requires targeted interventions to improve accessibility and availability, particularly in socioeconomically disadvantaged communities where adherence is often limited by financial and educational barriers. The Health Kiosk project addresses these challenges by promoting education and awareness in vulnerable populations. Understanding whether adherence to the Mediterranean diet is associated with improved health-related quality of life (HRQoL) in these groups is critical to informing effective public health strategies and tailoring interventions to overcome existing barriers within Portuguese neighborhoods.

## 2. Materials and Methods

### 2.1. Study Design and Sample

A cross-sectional analysis was performed among the citizens frequenting the Health Kiosk located at the Social Center of S. José (Bairro da Rosa, Coimbra) in Portugal. The “Health Kiosk” is a community center, created by the EU_SHAFE [18] (Europe enabling Smart Healthy Age-Friendly Environments) project in seven European cities, that enhances a “sharing learning” methodology between communities and a multidisciplinary team of health professionals. The aim was to prioritize education sessions to promote more sustainable environments within these communities and design multilevel policies in the future.

The focus of the study was to quantitatively measure the association between adherence to the Mediterranean diet and HRQoL scores and assess how lifestyle and demographic factors can impact that relationship. The study consisted of a planning phase (August 2022 to February 2023), with a first assessment through the application of data collection instruments, and an intervention phase (April 2023 to July 2023), where the educative sessions were held.

From the first assessment, 107 citizen users of the S. José Social Center showed interest in participating in the Health Kiosk study. A final sample of 104 participants that signed the informed consent, aged between 20 and 77 years (71 women and 33 men), enrolled in the study. This sample represents 12% of the total habitants of the two neighborhoods (869 habitants distributed per 367 households).

Information about age, sex, education, housing, work situation, marital status, smoking habits and alcohol consumption, perception of healthy eating, and access to a family doctor was collected with a general questionnaire before randomization. Only 60 participants accepted to have their anthropometric measures taken (weight, waist, hip, and arm circumference), and their data were used in the data analysis for these variables.

Body mass index (BMI) was calculated as kg/m^2^. Waist circumferences were measured according to the WHO expert consultation, in Geneve [19].

### 2.2. Data Collection—Questionnaires (SF-36 and MEDAS)

#### 2.2.1. MEDAS (Mediterranean Diet)

To assess adherence to the Mediterranean diet, the MEDAS questionnaire originally developed for the Spanish population [20] and later validated for the Portuguese population [21] was applied. This questionnaire consists of 14 questions related to the number of portions and frequency of consumption of typical (e.g., olive oil, nuts, fruits, vegetables, pulses, seafood) and non-typical Mediterranean foods (e.g., red or processed meats, sweetened beverages, and sweets, commercial bakery, or sugary desserts). Each item was scored as 0 or 1 depending on whether the criteria for adherence to the MedDiet was met. Adherence was then classified into three groups according to the score: <8 points indicate “poor adherence”; 9 to 10 points indicate “moderate adherence”; >11 to 14 points show a “high adherence” to the dietary pattern.

#### 2.2.2. SF-36 (Health-Related Quality of Life)

HRQoL was assessed using the European Quality of Life questionnaire (SF-36) [21,22,23], translated and validated to the Portuguese population [24]. This self-reported questionnaire is used to indicate the health status of populations, to help with service planning, and to measure the impact of clinical and social interventions. HRQoL questionnaire focuses on eight health concepts: physical functioning, bodily pain, general health perceptions, role limitations due to physical health problems, role limitations due to personal or emotional problems, emotional well-being, social functioning, and energy/fatigue. The first four can be grouped in the physical health domain and the other concepts in the social-emotional domain as represented by Ware et al. (Figure 1). It also includes an item that focuses on perceived change in health related to the past year. To score the questions we use the method from RAND 36-Item Health Survey 1.0 [25]. This questionnaire and methodology to calculate scores, could be accessed in the Appendix A.

A higher SF-36 score corresponds to a higher health-related quality of life.

### 2.3. Ethical Issues

The project was approved by the Ethical Commission of the Polytechnic Institute of Coimbra, with the number 169 CEIPC/2022. The principles of the Helsinki Declaration were respected and the participants in the study signed the informed consent after having the aims, objectives, and methods involved in the research study explained to them personally before the data collection.

### 2.4. Statistical Analysis

Baseline demographic characteristics were described as means (standard deviations) for quantitative traits and as proportions for qualitative traits through a descriptive analysis. According to the MEDAS questionnaire (with 14 items), after calculating the sum of total points, we categorized baseline adherence to the MedDiet into three groups “Low Adherence” (0–8 points), “Good Adherence” (9–10 points), and “High Adherence” (11–14). Quality of Life health concepts were categorized into two major dimensions: physical dimension and mental dimension.

Baseline characteristics and questionnaire mean scores were analyzed per MedDiet adherence group with median (IQR). For the Shapiro Walk Test, most of the variables were non-normally distributed. In that sense, for correlation analysis between the variables of interest (HRQoL health concepts and MedDiet adherence score) and baseline participant characteristics, the Spearman test was applied. Multiple linear correlation analysis was performed to understand the impact of HRQoL health concepts as predictors of change in MedDiet adherence.

The contingency table for categorical variables was used to show the relationship between baseline characteristics and MedDiet adherence score.

Independent variables of interest were age, gender, education, household, marital status, employment status, physical activity, smoking habits, alcohol consumption, “Considering having a healthy diet” and MedDiet score. Gender was dichotomized into (0) female and (1) male. Physical activity and smoking habits were dichotomized as to (1) Yes/No (0) answers. “Considering having a healthy diet” is divided into No/Maybe/Yes answers. Alcohol consumption is divided into three groups: <2 drinks per day, 2–4 drinks per day, and ≥5 drinks per day. Employment status is divided into five categories, unemployed (0), sick leave (1), self-employed (2), employee (3), and retired (4). Marital Status was divided into four groups: single (0), Married (1), Widowed/widower (2), and divorced (3). The household was divided into three groups: homeless (0), rented (1), own housing (2) and education into six groups: no education (0), 1st cycle (1), 2nd cycle (2), 3rd cycle (3), secondary education (4) and higher education (5). The influence of independent variables on quality of life were analyzed with multiple linear regression models. All the statistical analyses were performed with the software IBM^®^ SPSS Statistics 28 [26].

## 3. Results

### 3.1. Baseline Characteristics of Participants

Of the 107 interested individuals, 104 signed the informed consent to participate in the study and 102 responses to baseline questionnaires were considered eligible.

Participants were aged between 20 and 77 (mean 52.28 ± 14.0), mostly female (68.6%), unemployed (60.8%), and single (40.2%). The majority had a family doctor (88.2%) and had not reached secondary education (81.3%). The majority were non-smokers (69.6%), with 52.9% not practicing physical activity at all. Furthermore, 61.8% of the participants affirmed having a healthy diet. A total of 60 participants accepted to be screened for anthropometric measures (height; body weight; and arm, waist, and hip circumferences), and the results can be seen in Table 1.

The data were non-normally distributed for most of the variables, the health concepts of the SF-36 quality of life questionnaire, and the Mediterranean diet score.

### 3.2. Outcome Measures

#### 3.2.1. MedDiet Adherence Groups

Adherence to the MedDiet was tested with the MEDAS questionnaire, with 85% of the participants having a score within the category of “low adherence” (<8 points), of which 67.8% were female; 13% showing “moderate adherence” (9–10 points); and only 2% falling in the “high adherence” (>11–14 points) group, all of whom were female. The participants’ demographic and anthropometric characteristics were then analyzed within each MedDiet adherence group (Table 1).

No significant differences were seen between groups; however, higher values for the variables of body weight and BMI could be seen in the low adherence group compared to the other groups. Moreover, there was a higher prevalence of sedentary lifestyles among the low adherence group, with no daily practice of physical activity (Table 1).

#### 3.2.2. SF-36 Score Distribution Based on MedDiet Adherence Groups

The total SF-36 scores were shown to be similar between the low and moderate adherence groups and slightly better in the high adherence group. When comparing the eight health concepts within the physical and mental dimensions, the physical dimension seemed to have a linear association with adherence to the MedDiet, with the low adherence group having a mean of 236 points, the medium adherence group a mean of 254, and the high adherence group a mean of 275 points.

For the mental dimension, the results diverged between groups, with the low adherence group showing a higher score (262 points) compared to the moderate adherence group (241 points), almost equal to the high adherence group with 270 points. When individualizing by health concept, we can see that in the last four concepts, corresponding to the social–emotional domain, the lines for the high and moderate MedDiet adherence groups cross over the low adherence group more acutely in the energy/fatigue and social functioning health concepts (Figure 2). This variability can be explained by the fact that mental aspects are more subjective to analyze and can easily change from person to person.

However, the multiple linear regression analysis showed no significant association between MedDiet adherence score and HRQoL domains. The total physical domain score had a *p*-value of 0.885, and the social–emotional domain had a *p*-value of 0.573 in relation to Mediterranean diet adherence (Table 2).

Within a 95% confidence interval, no relationship was observed between these variables, as the lower and upper limits showed an almost zero change in score points.

### 3.3. Correlation Analysis Between the Variables of Interest

#### 3.3.1. Relation Between Quality of Life and Mediterranean Diet Adherence

We assessed the relation between HRQoL and MedDiet using Spearman correlation. No correlation between stool between MedDiet adherence and quality of life score was found (*p* = 0.83), both for physical (*p* = 0.44) and social-emotional domain (*p* = 0.55) (Table 3).

#### 3.3.2. Relation Between Quality of Life and Baseline Characteristics

Regarding participants’ characteristics, age was negatively correlated with the mental (rho = −0.23, *p* = 0.02)) and physical dimensions (rho = −0.41, *p* = <0.01) and with total SF-36 score (rho = −0.37, *p* = <0.01), showing that older people tend to have a worse quality of life. On the other hand, no correlation was seen with the MedDiet adherence score (rho = −0.03, *p* = 0.79). Daily practice of physical activity and having a higher education level (Figure 3) were statistically significantly correlated with quality of life (both total score and the two dimensions) (Table 2).

The energy and fatigue health concept showed a positive statistical correlation between physical activity status and perceived adherence to a healthy diet. A statistically significant negative correlation was found with anthropometric measures, including body weight and body circumferences (waist, arm, and hip), suggesting that excessive weight and larger body size may be associated with reduced vitality and energy in daily tasks.

Regarding the first assessment questionnaire, in response to the question “Do you consider that you have a healthy diet?”, there were statistically significant differences with the physical dimension score (rho = 0.197, *p* = 0.047) and the energy/fatigue health concept (rho = 0.210, *p* = 0.034), and almost significant differences with the total SF-36 score (rho = 0.19, *p* = 0.057).

#### 3.3.3. Relation Between Mediterranean Diet Adherence and Baseline Characteristics

Despite being weakly correlated, baseline body weight and body circumferences (waist, hip, and arm) showed a statistically significant negative association with the MedDiet adherence groups, which can reflect the importance of following good and healthy dietary patterns to keep a balanced body composition. Physical activity and daily alcohol consumption showed a weak but statistically significant correlation with a better adherence to the Mediterranean diet, which are principles of this healthy life pattern (Figure 4).

#### 3.3.4. Multivariable Logistic Regression Analysis

The variable of age was statistically significant (*p* ≤ 0.001), and it is estimated that when age increases by one year, the quality of life variable decreases by 6308 points (Table 4).

Below the 10% significance level, male gender was associated with an increase of 72.541 points in quality of life compared to being female (*p* = 0.073). Similarly, the factor “considering having a healthy diet” was associated with an increase of 50.361 points (*p* = 0.089).

Smoking habits, despite not being statistically significant (*p* = 0.4), were estimated to result in a decrease of 33.359 points in quality of life for people who smoke compared to those who do not smoke. Physical activity, however, was statistically significant (*p* = 0.002), with an estimated increase of 123.7 points in the quality of life score for those who practice physical activity daily.

The MedDiet score was estimated to lead to a modest decline in quality of life of −33.784 points for each one-unit increase on the MEDAS scale. However, that difference was not statistically significant (*p* = 0.434).

## 4. Discussion

This study highlights the low adherence to the Mediterranean diet (MedDiet) among individuals in socioeconomically disadvantaged groups, with only 2% demonstrating high adherence. This is significantly lower compared to previous national surveys, emphasizing the influence of income and education on dietary patterns [2,13,27,28]. The association between low socioeconomic status and reduced health-related quality of life (HRQoL) is well-documented [29], and this study reinforces the need to address systemic barriers that hinder MedDiet adherence, especially in southern European countries, where adherence has been declining among elderly and low-income populations [4,20,28,29,30]. Notably, the finding that 27.4% of participants had four or fewer years of education aligns with prior research indicating that lower education levels are associated with poor adherence to the MedDiet [14,30].

Our findings also revealed that low MedDiet adherence correlates with higher BMI and larger body circumferences, which can negatively impact health and quality of life. These results are consistent with other studies conducted in southern Europe that identified key determinants of MedDiet adherence and their relationship to body composition [5,30,31]. Waist circumference, a measure of visceral adiposity, is particularly significant, as it has been strongly associated with cardiovascular morbidity, all-cause mortality, and diminished quality of life [19,32,33].

Contrary to previous studies that identified positive correlations between MedDiet adherence and HRQoL [15,34,35], this study found no significant association between overall MedDiet adherence and HRQoL scores. However, specific HRQoL domains and lifestyle factors demonstrated significant relationships. Physical activity, for instance, was positively correlated with both MedDiet adherence and physical health dimensions of HRQoL, echoing findings from earlier research [16,31,36]. Interestingly, while prior studies linked MedDiet adherence to the mental dimension [31], this association was not confirmed in our study. Instead, we observed a negative relationship between the energy/fatigue component of HRQoL and anthropometric measures, such as arm, hip, and waist circumferences. Anthropometric measures, such as waist circumference and subcutaneous fat distribution using the hip circumference and arm circumferences, serve as cut-off points for central obesity and visceral fat. Values near the upper limits can suggest an overweight status, which can impact cardiometabolic health in the future [27].

Being overweight is related to lower vitality and energy levels in daily tasks, which possibly implies a lower quality of life in performing simple daily tasks such as lifting or carrying groceries, climbing several flights of stairs, or bathing and dressing by yourself. It can also affect one’s energy and mental/emotional drive to perform these tasks.

The variable “Considering having a healthy diet” was significantly associated with higher HRQoL scores, underscoring the potential role of self-perceived dietary habits in maintaining quality of life [37,38,39]. This indicates that fostering awareness and education about healthy eating could have a meaningful impact on HRQoL. However, effective interventions may require more personalized approaches that address the specific needs and challenges of individual families, rather than relying solely on generalized guidelines. This observation suggests that in these populations, factors such as income perception, social inclusion, and food security may play a more significant role in shaping HRQoL than dietary adherence alone [39,40,41].

This study has several limitations. Its cross-sectional design prevents the establishment of causality, and the overrepresentation of female participants reduces sample heterogeneity, potentially influencing the observed HRQoL correlations. The low percentage of participants with moderate or high MedDiet adherence (14.7%) further limits comparisons between adherence groups.

Another limitation is the alcohol consumption categorization, as non-drinkers were not considered individually. The lowest category, “less than 2 drinks per day”, also includes those who have only one drink per day, which can lead to a skewed and biased interpretation of the health outcomes.

Additionally, selection bias may have occurred as participants from the same households were included, reducing the sample’s diversity. Furthermore, the lack of data on disease status and household income introduces potential residual confounding. It is also debatable whether correlation analysis was the most appropriate statistical method for variables such as “Considering having a healthy diet”, which involve ordinal categorical responses.

Despite these limitations, this study has several strengths. As a pilot project in Portugal, it employed a novel share-methodology approach, fostering proximity with socioeconomically disadvantaged communities. The detailed descriptive analysis provides valuable insights into the barriers faced by these groups, offering a foundation for future policy development. Moreover, the extensive data collected during the first assessment enhance the study’s potential to inform multilevel policy interventions.

The lack of association between MedDiet adherence and HRQoL in this study suggests that the Mediterranean diet may not serve as a primary driver of quality of life in disadvantaged populations. Instead, broader socioeconomic factors, including income stability, social inclusion, and food security, likely play a more pivotal role. Future intervention studies should explore whether targeted education sessions and tailored approaches can improve MedDiet adherence and overall health outcomes in these communities.

## 5. Conclusions

This study underscores the low adherence to the Mediterranean diet (MedDiet) among socioeconomically disadvantaged populations, with significant barriers such as low income and education levels contributing to this trend. Although no direct association was found between MedDiet adherence and overall health-related quality of life (HRQoL), specific lifestyle factors, including physical activity and perceptions of healthy eating, were positively linked to HRQoL. These findings suggest that while the MedDiet alone may not be a primary driver of quality of life in these populations, its potential benefits could be amplified through targeted interventions addressing broader socioeconomic determinants.

The study highlights the importance of addressing factors such as food accessibility, social inclusion, and income stability, which likely play a more pivotal role in shaping HRQoL. Despite its limitations, including the cross-sectional design and limited diversity in adherence groups, the study provides valuable insights into the challenges faced by disadvantaged communities. Future longitudinal studies and intervention programs should focus on personalized and context-specific strategies to improve dietary habits and promote overall well-being in these vulnerable populations.

By addressing the systemic barriers that hinder adherence to healthy dietary patterns, public health initiatives can play a crucial role in fostering better health outcomes and quality of life in socioeconomically disadvantaged groups.

## Figures and Tables

**Figure 1 nutrients-17-00906-f001:**
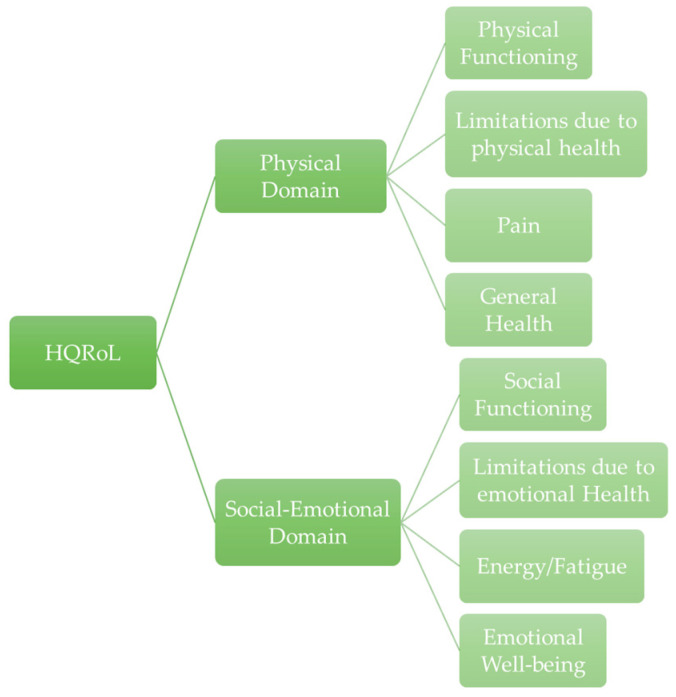
Health-Related Quality of Life Domains and Concepts (SF-36).

**Figure 2 nutrients-17-00906-f002:**
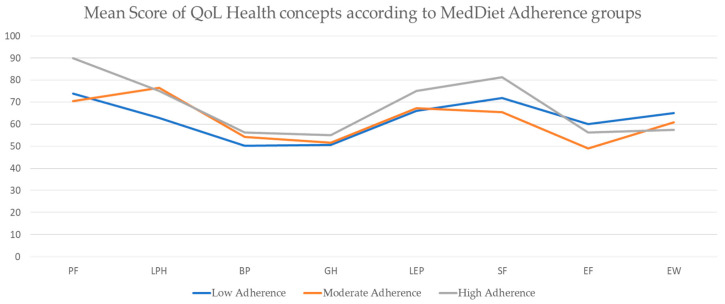
Quality of life health concepts (SF-36) according to the Mediterranean diet adherence groups (MEDAS). Abbreviations: PF, physical functioning; LPH, limitations due to physical health; BP, body pain; GH, general health; LEP, limitations due to emotional health; SF, social functioning; EF, energy and fatigue; EW, emotional well-being.

**Figure 3 nutrients-17-00906-f003:**
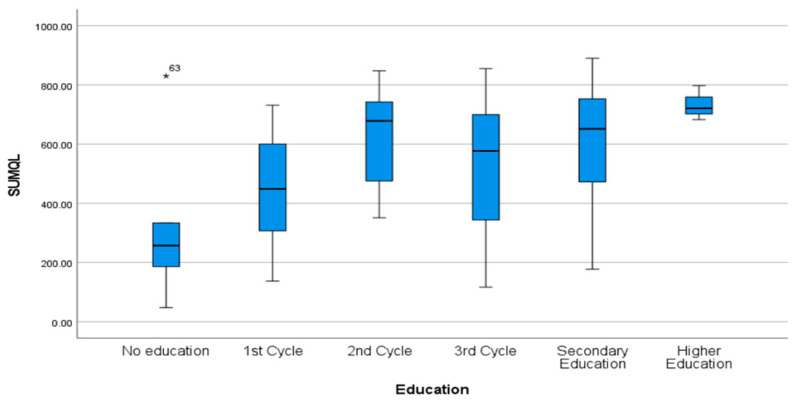
Box plot of total quality of life health score (SF-36) stratified by education level. Abbreviations: SUMQL, total quality of life score. * *p* > 0.05.

**Figure 4 nutrients-17-00906-f004:**
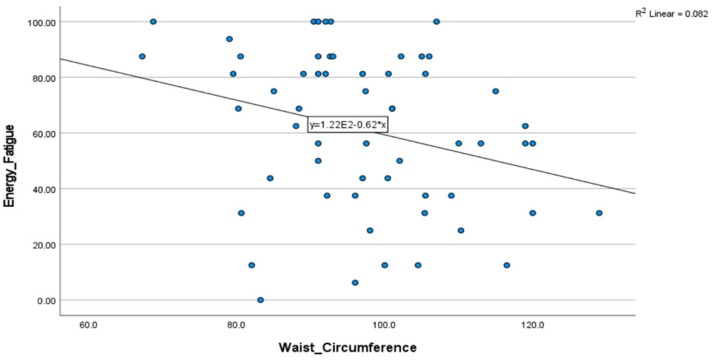
Scatter plot of the negative correlation between waist circumference and energy (SF-36 health concept). Abbreviations: SF-36, 36-item Short-Form Survey.

**Table 1 nutrients-17-00906-t001:** Baseline characteristics of the participants according to adherence to the Mediterranean diet.

	Low Adherence	Moderate Adherence	High Adherence
N *	87	13	2
MedDiet score, mean (SD)	6.44 (1.33)	9.38 (0.51)	11 (0.0)
SF-36 score, mean (SD)	545.99 (202)	541.25 (233.4)	608.75 (243.95)
Age (years), median (IQR)	55 (19)	59 (30)	49 (0)
Female gender, n (%)	59 (67.8%)	9 (69%)	2 (100%)
Marital status, n (%)			
Single	34 (39%)	7 (53.8%)	0
Married	22 (25.3%)	3 (23%)	1 (50%)
Widowed/widower	5 (5.7%)	1 (7.7%)	0
Divorced	26 (27.9%)	2 (15.4%)	1 (50%)
Level of education, n (%)			
No education	4 (4.6%)	1 (7.7%)	0
1st cycle	20 (23.0%)	3 (23%)	0
2nd cycle	28 (32.2%)	2 (15.4%)	1 (50%)
3rd cycle	21 (24.1%)	3 (23%)	0
Secondary education	12 (13.8%)	3 (23%)	1 (50%)
Higher education	2 (2.3%)	1 (7.7%)	0
Household, n (%)			
Homeless	8 (9.2%)	0	0
Rented	73 (83.9%)	12 (92.3%)	2 (100%)
Own housing	6 (6.9%)	1 (7.7%)	0
Work situation, n (%)			
Unemployed	51 (58.6%)	9 (69.2%)	2 (100%)
Sick leave	4 (4.6%)	0	0
Self-employed	9 (10.3%)	1 (7.7%)	0
Employee	2 (2.3%)	1 (7.7%)	0
Retired	21 (24.1%)	2 (15.4%)	0
Tobacco consumption, n (%)			
Non-smoker	61 (70.1%)	8 (61.5%)	2 (100%)
Smoker	26 (29.9%)	5 (38.5%)	0
Physical activity, n (%)			
None	49 (56.3%)	5 (38.5%)	0
<30 min per day	10 (11.5%)	0	0
≥30 min per day	28 (32.2%)	8 (61.5%)	2 (100%)
Alcohol consumption, n (%)			
<2 drinks per day	86 (98.9%)	11 (84.6%)	2 (100%)
2–4 drinks per day	0	2 (15.4%)	0
≥5 drinks per day	1 (1.1%)	0	0
Do you consider that you have a healthy diet? n (%)			
No	20 (23.0%)	0	0
Yes	50 (57.5%)	11 (84.6%)	2 (100%)
Maybe	17 (19.5%)	2 (15.4%)	0
Body weight **, median (IQR)	80.95 (27.6)	66.85 (10.9)	72.25 (-)
BMI **, median (IQR)	30.14 (9.45)	25.9 (6.4)	29.45 (-)
Arm circumference (cm) **, median (IQR)	34.25 (5.7)	32 (4.4)	32 (-)
Waist circumference (cm) **, median (IQR)	97.2 (14.9)	91.85 (15.1)	92.75 (-)
Hip circumference (cm) **, median (IQR)	105 (16)	100.45 (15.4)	109.7 (-)

Abbreviations: MedDiet, Mediterranean diet; SF-36, 36-item Short-Form Survey; * N = 102; ** n = 60.

**Table 2 nutrients-17-00906-t002:** Beta coefficients of multiple linear regression between HRQoL domains and MedDiet adherence.

Model	Unstandardized Coefficients		Standardized Coefficients	t	Sig.	95% Confidence Interval for B	
	B	Std. Error	Beta			Lower Bound	Upper Bound
SUMPD	0.000	0.002	0.020	0.145	0.885	−0.004	0.005
SUMSED	−0.001	0.002	−0.076	−0.566	0.573	−0.006	0.003

Dependent variable: SUMMED. Abbreviations: SUMMED, Mediterranean diet score; SUMPD, total physical domain score; SUMSED, total social–emotional domain score.

**Table 3 nutrients-17-00906-t003:** Bivariate correlations between Mediterranean diet adherence score, health-related quality of life, and lifestyle.

Spearman Correlation					
Variables	MedDiet Adherence Score (MEDAS)	Total Score (SF-36)	Social–Emotional Domain (SF-36)	Energy/Fatigue, Median (IQR)	Physical Domain (SF-36)
MedDiet Adherence	1	0.21 (0.832)	−0.06 (0.547)	−0.126 (0.205)	0.08 (0.444)
Health concepts					
Energy/fatigue, median (IQR)	−0.13 (0.205)	0.78 (<0.001)	0.80 (<0.001) **	1	0.62 (<0.001) **
Participant characteristics					
Physical activity	0.22 (0.029) *	0.361 (<0.01)	0.28 (<0.01) **	0.26 (<0.01) **	0.345 (<0.01) **
Healthy diet	0.095 (0.34)	0.189 (0.057)	0.177 (0.075)	0.210 (0.034) *	0.197 (0.047) *
Alcohol consumption	0.25 (0.01)	0.13 (0.198)	0.15 (0.128)	0.09(0.39)	0.1 (0.294)
Education	0.1 (0.34)	0.28 (<0.01)	0.216 (0.03) *	0.19(0.05) *	0.28 (<0.01)
Body weight	−0.25 (0.057)	−0.12 (0.4)	−0.14 (0.29)	−0.25(0.05) *	−0.07 (0.5)
Body circumferences					
Waist circumference (cm)	−0.22 (0.09)	−0.15 (0.25)	−0.19 (0.2)	−0.3 (0.02) *	−0.08 (0.83)
Hip circumference (cm)	−0.07 (0.59)	−0.14 (0.29)	−0.23 (0.08)	−0.31 (0.02) *	−0.03 (0.82)
Arm circumference (cm) **	−0.22 (0.07)	−0.16 (0.2)	−0.19 (0.15)	−0.3 (0.02) *	−0.11 (0.39)

Abbreviations: MedDiet, Mediterranean diet; SF-36, 36-item Short-Form Survey; * *p*-value = 0.05; ** *p*-value < 0.01.

**Table 4 nutrients-17-00906-t004:** Beta coefficients of multiple linear regression between influencing factors and HRQoL.

Influencing Factors (Model)	Unstandardized B	SE	*p*-Value
Constant	685.332	113.583	<0.001 **
Age	−0.308	1.753	<0.001 **
Gender	72.541	40.047	**0.073**
Education	17.402	17.394	0.320
Household	54.668	49.669	0.274
Marital status	2.494	16.047	0.877
Employment status	17.206	12.120	0.159
Physical activity	123.701	39.263	0.002 **
Smoking habits	−33.359	39.453	0.400
Alcohol consumption	25.187	79.073	0.751
Healthy eating	50.361	29.314	**0.089**
MedDiet Score	−33.784	43.013	0.434

Abbreviations: HRQoL, health-related quality of life; MedDiet Score, Mediterranean diet score. ** *p*-value < 0.01. Bold values are below 10% significance.

## Data Availability

The original contributions presented in this study are included in the article/Appendix A. Further inquiries can be directed to the corresponding author.

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
