# Peer review of "Low Mediterranean Diet Adherence Is Associated with Poor Socioeconomic Status and Quality of Life: A Cross-Sectional Analysis"

_nutrients, 2025, doi:10.3390/nu17050906_

Round 1
Reviewer 1 Report
Comments and Suggestions for Authors
Evaluation of manuscript Nutrients-3439022
This is an interesting manuscript about mediterranean diet and health, I will present below my consideration about the text aiming help the authors.
You should not use acronyms (EU_SHAFE) in the title, please review.
The abstract needs to improve the quality of the data, the authors explore secondary data, however, there is much more important data in Tables 2 and 3 and Figures 3 to 6. Please review.
Please review the introduction, page 2 line 52 – insert the reference.
The first time you create an Acronyms, use it from this point forward. The Acronyms MedDiet is used several times throughout the text. Furthermore, the authors created another PREDIMED acronym. Finally, I emphasize that it is correct to present the full term followed by (acronyms), review page 1 line 23 and 24 [EU_SHAFE (Europe enabling Smart Healthy Age-Friendly Environments)]. The same can be seen for QoL. Please review the entire text.
In the introduction, the authors state that there is a decrease in adherence to the Mediterranean diet, without presenting a plausible justification for this, so I suggest that they combine the 3rd and 4th paragraph on page. 2. Not only at this point, but there are many fragmented ideas in the introduction, see, the 5th and 6th paragraph are continuity of each other, they must be unified. Please review.
At the end of the introduction, the justification for the study is not clear. I believe the authors need to explain what the Health Kiosk project is.
Regarding the methods, the authors present that the sample size represents 12% of the population. However, it is important to show other sociodemographic data (education, social level, occupation) that show the representativeness of the population.
Due to the low sample size, I suggest that participants with moderate and high adherence to the Mediterranean diet are a single group. Please redo the analyses and rewrite the results.
In the discussion there are results that are repeated, in the discussion the authors must base themselves on works already published and explain the results, please review. Furthermore, there are loose texts in the discussion, such as lines 313 to 315 on page. 11. I believe that the main limiting factor of the present study is the low sample representation of the High adherence group (n=2). Therefore, I recommend redoing the statistical calculations grouping moderate and high and redoing the discussion based on the new results.
Author Response
Dear review,
Thank you very much for your comments, that contributes to the improvement of the quality of our paper.
Bellow, we answer your comments.
You should not use acronyms (EU_SHAFE) in the title, please review.
EU SHAFE acronym was removed from the title.
The abstract needs to improve the quality of the data, the authors explore secondary data, however, there is much more important data in Tables 2 and 3 and Figures 3 to 6. Please review.
Abstract was reformulated to emphasize these results.
Please review the introduction, page 2 line 52 – insert the reference.
Reference added.
The first time you create an Acronyms, use it from this point forward. The Acronyms MedDiet is used several times throughout the text. Furthermore, the authors created another PREDIMED acronym. Finally, I emphasize that it is correct to present the full term followed by (acronyms), review page 1 line 23 and 24 [EU_SHAFE (Europe enabling Smart Healthy Age-Friendly Environments)]. The same can be seen for QoL. Please review the entire text.
Revised.
In the introduction, the authors state that there is a decrease in adherence to the Mediterranean diet, without presenting a plausible justification for this, so I suggest that they combine the 3rd and 4th paragraph on page. 2. Not only at this point, but there are many fragmented ideas in the introduction, see, the 5th and 6th paragraph are continuity of each other, they must be unified. Please review.
Revised.
At the end of the introduction, the justification for the study is not clear. I believe the authors need to explain what the Health Kiosk project is.
In our opinion the reference to Health Kiosk in Introduction/justification could be confused, reason why, was removed. However, the justification of the study was improved.
Regarding the methods, the authors present that the sample size represents 12% of the population. However, it is important to show other sociodemographic data (education, social level, occupation) that show the representativeness of the population.
“This sample represents 12% of the total habitants of the two neighborhoods (869 habitants distributed per 367 households).”
Related to this sentence, I would like to explain that this study analyse the participants in Health Kiosk in a sample of N=104 from a population of N=107. The information that we give about neighborhoods were the kiosk was implemented in to characterize the intervention. However we can remove this information, because, maybe is confused. We describe the sociodemographic chacteristics:
“Participants were aged between 20 and 77 (mean 52.28 ± 14.0), mostly females (68.6%) and unemployed (60.8%) and single (40.2%), with family doctor (88.2%) and the majority having not reach secondary education (81.3%). The majority were non-smokers (69.6%) with 52.9% not practicing physical activity at all. 61.8% of the participants affirm to have a healthy diet.”
Due to the low sample size, I suggest that participants with moderate and high adherence to the Mediterranean diet are a single group. Please redo the analyses and rewrite the results.
Considering the information/answer above, we decided to purpose that the fusion of these two groups not be considered.
In the discussion there are results that are repeated, in the discussion the authors must base themselves on works already published and explain the results, please review. Furthermore, there are loose texts in the discussion, such as lines 313 to 315 on page. 11. I believe that the main limiting factor of the present study is the low sample representation of the High adherence group (n=2). Therefore, I recommend redoing the statistical calculations grouping moderate and high and redoing the discussion based on the new results.
Discussion was revised, but not the recalculation of the groups.
We really understand the results, considering the group of disadvantage people, their food habits and the criteria of mediterranean diet adherence.
Reviewer 2 Report
Comments and Suggestions for Authors
-
Invalid research group: Individuals with lower wealth pay less attention to their diet compared to wealthier individuals, which is due to their limited means.
-
Poor diet choice: The Mediterranean diet includes high-quality red wine, feta cheese (made from sheep's and goat's milk), nuts, and high-quality vegetable oils—products that are relatively expensive.
-
The comparison between individuals with lower education and adherence to this diet is incomprehensible: In the age of the Internet, everyone can access information about diets.
-
Lack of detailed income data: Each person might define their financial status differently.
-
Significant predominance of women (who statistically earn less than men): There is too much focus on women in the study.
-
The survey used to select the study group is poorly constructed: Access to a general practitioner, smoking, or alcohol consumption does not necessarily impact quality of life or financial status. There should be more questions about nutrition: the number of meal portions consumed and whether the person enjoys products characteristic of a particular diet.
-
Of 104 people, only 60 underwent measurements: These results are therefore unreliable. It is unclear which participants (age and weight) underwent the measurements.
-
Women aged 52 years (±14 years): What about younger individuals who have better access to education and more opportunities to seek information?
-
The majority of study participants were unemployed: It is evident that these individuals will have low financial status. The study should have included employed individuals (with stated income ranges).
-
Most participants did not adhere to the Mediterranean diet, which was associated with a lower standard and quality of life: This conclusion is unclear. There should be a comparison of at least two diets (preferably with similar costs), and then the quality of life should be assessed.
-
The conclusions could be expanded with specific results: There is too much general information. The last two paragraphs are not very relevant to the article's topic.
-
The article talks about the relationship between the quantitative consumption of products according to the Mediterranean diet model and demographic factors, social status. The aim is to discover this relationship through the Predimed and SF36 questionnaires, which is divided into two parts. The first deals with so-called physical issues and the second touches on psychological topics. The assumptions of the research design are the few plus points of this publication.
-
Research design: The participant group is predominantly female, with an equal number of men missing. The large age discrepancy will make the answers very ambiguous. An important issue in the study is the location. It is located at the S. José in Coimbra, Portugal, it has a health kiosk that supports the local community. This kiosk is part of the social and support services offered by Caritas Diocesana de Coimbra. It helps to combine health and social services, offering support to vulnerable adults, including the long-term unemployed and beneficiaries of social inclusion income. The place should be accessible to the general population and not where only people in need with low financial and health status come to.
-
Additionally, questionnaires have their drawbacks. Predimed - Participants may assess their eating habits based on subjective feelings, which can lead to misinformation. Participants may have difficulty remembering their eating habits accurately, which affects the reliability of the results. The survey may be distorted by other factors affecting health, such as lifestyle changes, which are not included in the questionnaire.
-
The SF-36 questionnaire is complex and may be difficult for some respondents to understand, which may affect the accuracy of responses. Due to the number of questions and variety of topics, the questionnaire may require more time to complete, which may lead to respondent fatigue.The SF-36 may not take into account cultural differences in perceptions of health and quality of life, which may affect results in different populations.
-
The manuscript is not clearly structured. Please elaborate on the topic and relevant references to the literature review in the theoretical section outlining the roles of the Mediterranean diet in other countries. There is no overview of the economic situation of the Portuguese population in the theoretical part.
-
The results are presented inaccurately and chaotically. The tables and figures have different styles. There is a lack of unification in the tables.
-
The study design cannot fully objectively answer the vague hypotheses set.
-
The conclusions are presented very incompletely through a very incomplete methodology.
General Assessment:
-
The study analyzes the relationship between adherence to the Mediterranean diet and health-related quality of life (HRQoL) among individuals from two disadvantaged neighborhoods in Portugal. While the topic is relevant, and the study could provide valuable insights into the impact of the Mediterranean diet on quality of life in these groups, there are several important shortcomings that need to be addressed.
Specific comments:
1. Line 195 – Lack of Age Group Categorization
-
- The authors did not divide the study participants into age subgroups, and the analysis was based on a group aged 20–77 years. There are significant differences between individuals in different age groups, so they cannot be treated as a homogeneous group. The lack of such a division may have affected the distortion of the results obtained.
2. Line 210 – Lack of a Representative Group of Mediterranean Diet Adherents
-
- Only 15% of the participants in the study adhered to the Mediterranean diet, and only 2% strictly followed it. Due to such a small number of individuals following this diet, comparing their results with those of 85% of participants who did not follow it is not valid and may lead to erroneous conclusions that do not reflect the actual impact of the Mediterranean diet on the aspects studied.
3. Line 210 – Lack of Data on the Duration of Mediterranean Diet Adherence
-
- The authors do not provide information on the duration for which individuals strictly adhered to the principles of the Mediterranean diet. If such data are available, they should be included in the article. If they were not collected, this issue should be noted and considered in future studies. The impact of the Mediterranean diet depends on the length of time the principles have been followed, which makes this information crucial when comparing the results of the adherent and non-adherent groups.
4. Line 342 – Failure to Consider Participants' Health Status
-
- In formulating conclusions, the authors did not consider the health status of the participants, which is a significant oversight. Illnesses have a substantial impact on quality of life, physical activity, and mental well-being, especially in older age groups. Omitting such an important factor in studies aiming to determine the relationship between quality of life and other variables may significantly distort the results.
Conclusion
-
Although the article addresses an important issue, there are several significant gaps that require attention. Categorizing age groups, improving the representativeness of the Mediterranean diet group, including information about the duration of diet adherence, and considering the participants' health status are key elements that should be addressed to increase the accuracy and reliability of the study's results. The authors should take these suggestions into account to enhance the transparency, validity, and value of the article.
-
This review points out the necessary changes and presents a critical perspective on the article.
-
Lines 42 to 46 – lack of citation for the claim that "50% of the Portuguese population exhibits overweight/obesity characteristics."
-
Methodologically, this study is sound, as two well-known validated questionnaires were used. However, the small sample size and the generalization of conclusions based on responses from 102 questionnaires (~70% men, ~30% women) to the entire Portuguese population is problematic. In the materials and methods section, it is stated that the group represents 12% of residents from two districts.
-
Only 60 out of 102 individuals agreed to anthropometric measurements. This limits the scope of inference.
-
The study participants were aged between 20 and 77 years, with an average age of approximately 52 years. This is quite a wide age range.
-
Correlation analysis was used to study qualitative variables, such as the perception of diet. This could be considered debatable.
-
No correlation between the Mediterranean Diet (MedDiet) and Health-Related Quality of Life (HRQoL) – the results may suggest that other factors, such as income or social inclusion, are more important for quality of life than diet itself. The small sample size with low adherence to the diet (2%) limits the conclusions.
-
The results are not entirely clear due to the lack of consideration for actual income levels when discussing the relationship between socio-economic status and adherence to the Mediterranean Diet (MedDiet).
Author Response
Dear review,
We appreciate your comments and contributions to our research. However, their are variables of the research that we can't control, because this is an observational study. Below we answer to your comments.
We hope that you understand our point of view, and consider that this paper should be a small contribution to the knowledge about MD and QoL in this scenarios of social disadvantage.
The Mediterranean diet isn't necessarily more expensive; local products, poultry meat, etc. and present a good cost-effectiveness.
https://pmc.ncbi.nlm.nih.gov/articles/PMC11206702/
https://www.unisa.edu.au/media-centre/Releases/2023/the-mediterranean-diet-good-for-your-health-and-your-hip-pocket/
The comparison between individuals with lower education and adherence to this diet is incomprehensible: In the age of the Internet, everyone can access information about diets.
We understand. However, to be in Internet don't mean be correct. Missinformation is a huge problem, mainly in individuals with lower education.
Lack of detailed income data: Each person might define their financial status differently.
I don't know about which variable is this comment.
Significant predominance of women (who statistically earn less than men): There is too much focus on women in the study.
We understand, but was observational, and the sample have more women.
The survey used to select the study group is poorly constructed: Access to a general practitioner, smoking, or alcohol consumption does not necessarily impact quality of life or financial status. There should be more questions about nutrition: the number of meal portions consumed and whether the person enjoys products characteristic of a particular diet.
It was a option just to analyse MDiet adherence. If future studies we will consider this.
Of 104 people, only 60 underwent measurements: These results are therefore unreliable. It is unclear which participants (age and weight) underwent the measurements.
Many people in the group are suspicious and have little self-confidence, which may have led them to not authorize anthropometric specifications to be made. Although the sample of individuals with anthropometric measurements is smaller than the general sample, we consider that we enriched the article, which is why we included it.
Women aged 52 years (±14 years): What about younger individuals who have better access to education and more opportunities to seek information?
They don't participate of the study. Maybe because are working and are not participants from health kiosk.
The majority of study participants were unemployed: It is evident that these individuals will have low financial status. The study should have included employed individuals (with stated income ranges).
Most of the residents in this neighborhood and that are supported by health kiosk are "chronic" unemployed. However, they are part of the society and need to be "studied" also.
Most participants did not adhere to the Mediterranean diet, which was associated with a lower standard and quality of life: This conclusion is unclear. There should be a comparison of at least two diets (preferably with similar costs), and then the quality of life should be assessed.
This is an observational study that shows that people with low adherence to MD have low quality of life.
The conclusions could be expanded with specific results: There is too much general information. The last two paragraphs are not very relevant to the article's topic.
Conclusion was revised.
The other comments were also considered in the revision.
Round 2
Reviewer 1 Report
Comments and Suggestions for Authors
Analyzing the second version of the manuscript, I see that the authors have improved a lot and satisfied the reviewers. However, I have one last caveat : Figures 2 and 3 are unnecessary. They can easily be replaced with text.
Author Response
Thank you very much.
We revised the manuscript and removed Figures 2 and 3.
Reviewer 2 Report
Comments and Suggestions for Authors
Review of the Manuscript
The authors have made some improvements to their manuscript, particularly in acknowledging the role of mental health at the beginning. However, the revisions are still insufficient, and significant concerns remain unaddressed.
- Mental Health Assessment: While the authors mention that the SF-36 questionnaire was used to evaluate mental and emotional well-being, there are no results presented for these aspects. If these variables were measured, the results should be clearly stated. Otherwise, referencing them without presenting data suggests an incomplete analysis. Without concrete results, the claim that mental health was assessed is misleading.
- Lack of Explanation for Key Variables: The study compares Mediterranean diet adherence with various health-related quality of life (HRQoL) components, such as emotional well-being and social functioning. However, there is no clear definition or explanation of what specific questions were asked regarding these factors. Given that the questionnaire was shortened, it is even more crucial to clarify what was included and what was omitted. Readers should not be left guessing what the study actually measured.
- Alcohol Consumption Categorization: The classification of alcohol intake is problematic. The lowest category is "less than 2 drinks per day," which fails to capture individuals who do not drink alcohol at all. This limitation could skew results, as non-drinkers and occasional drinkers might exhibit different health outcomes compared to those consuming alcohol daily. A more nuanced categorization, including a "non-drinker" category, would provide a clearer picture.
- Inconsistent Presentation of Results: The authors provide visual comparisons of Mediterranean diet adherence and HRQoL variables, but the text lacks clarity regarding the interpretation of these results. For example, in the discussion, they highlight a correlation between energy levels and body composition, yet there is no deeper exploration of what this means in practical terms. The study should explicitly connect findings to their potential implications.
- Unclear Statistical Reporting: The statistical analysis section describes various correlations, but some claims in the discussion lack sufficient statistical support. For example, it is stated that the mental health dimension results "diverge between groups," yet no clear explanation is given as to why or how significant these differences are. Including effect sizes or confidence intervals would enhance transparency.
Overall, while the manuscript has improved, it still lacks rigor in presenting mental health results, defining key variables, and structuring the statistical analysis. A more thorough revision is needed.
Author Response
Thank you for your comments.
We answered them bellow:
1)Mental Health Assessment: While the authors mention that the SF-36 questionnaire was used to evaluate mental and emotional well-being, there are no results presented for these aspects. If these variables were measured, the results should be clearly stated. Otherwise, referencing them without presenting data suggests an incomplete analysis. Without concrete results, the claim that mental health was assessed is misleading.
- SF 36 is a self-reported questionnaire used to indicate the health status of populations, that focuses on two main domains: physical and emotional.
- Figure 4 presents the mean scores of QoL concepts analyzed in the questionnaire according to the MedDiet adherence groups.
2)Lack of Explanation for Key Variables: The study compares Mediterranean diet adherence with various health-related quality of life (HRQoL) components, such as emotional well-being and social functioning. However, there is no clear definition or explanation of what specific questions were asked regarding these factors. Given that the questionnaire was shortened, it is even more crucial to clarify what was included and what was omitted. Readers should not be left guessing what the study actually measured.
- We added in the appendix the original questionnaire where we based the translated version and the scoring system (RAND 36-Item Health Survey 1.0- reference (25)) that we use in the different concepts from which domain.
3)Alcohol Consumption Categorization: The classification of alcohol intake is problematic. The lowest category is "less than 2 drinks per day," which fails to capture individuals who do not drink alcohol at all. This limitation could skew results, as non-drinkers and occasional drinkers might exhibit different health outcomes compared to those consuming alcohol daily. A more nuanced categorization, including a "non-drinker" category, would provide a clearer picture.
- We manifest that it’s a limitation of the study to not exhibit the non-drinkers category.
-In Portugal most of the man consume of alcohol< or = 2 drinks and there is consensual agreement that this amount is acceptable and according to the OMS health guidelines. Nevertheless, alcohol impacts HRQL and it could have made a difference in outcomes and as a confounding factor.
4)Inconsistent Presentation of Results: The authors provide visual comparisons of Mediterranean diet adherence and HRQoL variables, but the text lacks clarity regarding the interpretation of these results. For example, in the discussion, they highlight a correlation between energy levels and body composition, yet there is no deeper exploration of what this means in practical terms. The study should explicitly connect findings to their potential implications.
- We understand the possible lack of clarity on the explanation given to the matter, so we improved the text to a better understanding. As energy and fatigue are in the mental domain category but in our interpretation can also impact the drive to do daily physical tasks.
5)Unclear Statistical Reporting: The statistical analysis section describes various correlations, but some claims in the discussion lack sufficient statistical support. For example, it is stated that the mental health dimension results "diverge between groups," yet no clear explanation is given as to why or how significant these differences are. Including effect sizes or confidence intervals would enhance transparency.
- We improve the results text and added a new table of betta coefficients in linear regression analyses between the domains and med Diet adherence groups clearing the lack of statistically significant relationship between the variables.